# Effect of Dietary Brown Seaweed (*Macrocystis pyrifera)* Additive on Meat Quality and Nutrient Composition of Fattening Pigs

**DOI:** 10.3390/foods10081720

**Published:** 2021-07-26

**Authors:** Nancy Jerez-Timaure, Melissa Sánchez-Hidalgo, Rubén Pulido, Jonathan Mendoza

**Affiliations:** 1Instituto de Ciencia Animal, Facultad de Ciencias Veterinarias, Universidad Austral de Chile, Valdivia 5090000, Chile; rpulido@uach.cl; 2Escuela de Graduados, Facultad de Ciencias Veterinarias, Universidad Austral de Chile, Valdivia 5090000, Chile; melissaruth47@gmail.com; 3I+D Patagonia Biotecnología SA., Valdivia 5090000, Chile; jonathan.mendoza.mv@gmail.com

**Keywords:** pig, seaweed, pork quality, fatty acids, minerals, proximal composition, *Macrocystis pyrifera*

## Abstract

The objective of this study was to evaluate the effects of dietary brown seaweed (*Macrocystis pyrifera*) additive (SWA) on meat quality and nutrient composition of commercial fattening pigs. The treatments were: Regular diet with 0% inclusion of SWA (CON); Regular diet with 2% SWA (2%-SWA); Regular diet with 4% SWA (4%-SWA). After slaughtering, five carcasses from each group were selected, and *longissimus lumborum* (LL) samples were taken for meat quality and chemical composition analysis. Meat quality traits (except redness intensity) were not affected (*p* > 0.05) by treatments. Samples from the 4%-SWA treatment showed the lowest a value than those from the 2%-SWA and CON treatments (*p* = 0.05). Meat samples from the 4%-SWA group contained 3.37 and 3.81 mg/100 g more of muscle cholesterol than CON and 2% SWA groups, respectively (*p* < 0.05). The SWA treatments affected (*p* ≤ 0.05) the content of ash, Mn, Fe, and Cu. The LL samples from 4%-SWA had the highest content of ash; however, they showed 0.13, 0.45, and 0.23 less mg/100 g of Mn, Fe, and Zn, respectively, compared to samples from CON (*p* ≤ 0.05). Fatty acids composition and macro minerals content (Na, Mg, and K) did not show variation due to the SWA treatments. Further studies are needed to understand the biological effects of these components on adipogenesis, cholesterol metabolism, and mineral deposition in muscle.

## 1. Introduction

Pork meat is generally recognized as an excellent source of proteins, B-vitamins, minerals, especially heme iron, trace elements, and other bioactive compounds [1]. Pork has been the most widely consumed meat in the world up to 2018, accounting for 40.1% of the total global meat intake [2]. The nutritional composition of meat and its organoleptic attributes depends on many factors such as genetics, age, sex condition, production system, diet, and location of cuts/muscles [3]. Pork quality has been the primary concern for producers, researchers, meat packers, processors, retailers, and ultimate consumers [4]. However, in the last decades, consumers have been demanding healthier and nutritious foods [5]. This demand has motivated a growing interest in the study of natural supplements in pig nutrition that could enhance the productive performance and health of the animals but also improve meat quality and nutrient composition.

Currently, there is a worldwide increase in the use of seaweed as a supplement or additive in animal production, especially on sustainable production systems [6,7]. Due to the increasing interest in organic foods, seaweed represents an alternative to avoid chemical or synthetic ingredients in the diet of animals [7,8].

Among the huge variety of seaweeds, brown seaweed is widely distributed throughout the globe and is also very abundant on the Chilean coast [9]. There are approximately 1500–2000 species of brown algae worldwide, and some species, such as *Macrocystis pyrifera* (giant kelp), play an important role in the ecosystem, growing up to 20 m and forming underwater kelp forests [10]. Their big size and easy harvesting management offer important advantages for animal consumption [11].

In general, brown seaweed has a lower protein content and lipids, but it contains higher levels of minerals, fatty acids, and essential amino acids and bioactive components than red and green seaweeds [11]. Moreover, brown seaweed contains vitamins A, B1, B12, D, E, and C, and folic acid, riboflavin, niacin, and pantothenic acid [12] and are an excellent source of antioxidants and bioactive compounds [13,14].

The inclusion of seaweed additives in the diet of lambs [15], beef cattle [16], fish [17], laying hens [18], and rabbits [19] has been studied. Additionally, the use of seaweed extracts in growing and fattening pigs has shown an improved growth performance and feed efficiency [9,20,21,22]. Extensive reviews about the inclusion of seaweeds in monogastric production have been published [21,23,24]. However, fewer studies have investigated the effects of dietary seaweed on pork carcass/meat quality traits [22,25,26,27,28]; but none of them has reported the effects of these natural additives on pork meat composition and quality.

The magnitude of the associate response to the inclusion of seaweed in the animals’ diet on growth performance and carcass traits depends on the type of seaweed used (red, brown, green, and yellow), the bioactive components present in the extract, and the proportion and frequency used in the diet [11]. The objective of this study is to evaluate the effects of the inclusion of brown seaweed (*Macrocystis pyrifera*) additive in the diet of fattening pigs on pork meat quality and nutrient composition.

## 2. Materials and Methods

### 2.1. Sampling and Seaweed Additive Description

The study was conducted with 240 pigs (castrated males and females) of approximately 14 weeks of age (52.5 ± 2.8 kg) in an intensive production unit located in the Ñuble Region (Chile). Before the fattening phase, animals were separated into 12 groups of 20 pigs each. Groups were assigned to one of the following treatments: Control Group: Regular diet with 0% of seaweed additive (SWA); 2%-SWA: Regular diet + 20 kg of SWA per 1000 kg of concentrate; 4%-SWA: Regular diet + 40 kg of SWA per 1000 kg of concentrate. Pigs were balanced by weight and sex condition. The SWA is a lyophilized product of *Macrocystis pyrifera*, that maintains its chemical-physical characteristics and bioactive compounds. The composition of the regular diet and the chemical composition of the SWA are shown in Table 1.

Five female pigs per treatment/replicate were selected for the *postmortem* evaluation to avoid the sex variation factor. The average final live weight was 106.7 ± 2.2 kg. The animals were transported to a slaughterhouse plant facility located 30 km away from the commercial farm and slaughtered after 12 h of lairage. The carcasses were chilled for 48 h *postmortem* at 4 °C. The entire portion of the *longissimus lumborum* (LL) muscle was removed from each left carcass side, and samples of 2.5 cm thickness were obtained. Two samples were used immediately for pH and color evaluation, and four samples of each loin were packaged and frozen at −20 °C for 30 days for the rest of the analysis.

### 2.2. Meat Quality Evaluation

Instrumental color was measured in fresh samples 48 h *postmortem*. A Hunter Lab Mini Scan XE Plus (Hunter Associates, Reston, VA, USA) was used with a 2.5-cm open port, Illuminant A, and 2° standard observer to objectively evaluate color. Three readings were obtained from the muscle surface, and the mean was calculated. Readings were obtained after exposing the muscles to air for 30 min (bloom). The color scale used was Hunter L, a, b. The L value represents lightness; a and b values represent redness, and yellowness, respectively. Warner–Bratzler Shear force (WBSF) and Water Holding Capacity (WHC) were evaluated in samples cooked in a convection oven (Albin Trotter model E-EMB Digital) to a final internal temperature of 70 °C following the guidelines of the American Meat Science Association [29]. The temperature was monitored using an Omega thermocouple thermometer type T (Omega Engineering, Inc., Stamford, CT, USA) inserted into the geometric center of each steak. The cooked steaks were chilled for 2 h at 2 °C, and then eight cores (1.27 cm in diameter) were removed parallel to the muscle fiber orientation. Cores were sheared once each on the Warner-Bratzler Meat Shear apparatus (GR Manufacturing Co., Manhattan, NY, USA) to get WBSF values. WHC was determined as cooking loss, which was determined by weight, expressed as a percentage compared to the original weight of the sample. A taste preference test was performed with 19 panelists (16 women and 3 men). Two steaks of each treatment were used in each session (two sessions). The tests were carried out in individual evaluation cabinets illuminated with red light. Each panelist, in each session, tasted three samples (one from each treatment) at random, and they were asked to select the best preference of the three samples.

### 2.3. Nutrient Composition of Meat

Moisture, protein, and fat content of meat samples were determined according to the AOAC [30]. All experiments were done in triplicate. Duplicates of 10 g of ground meat were calcined in a furnace at 550 °C for 6 h. After cooling, the residue (white ash) was subjected to an acid digestion process with 10 mL of a 20% *v*/*v* hydrochloric acid solution by heating on a hot plate for 10 min. Mineral analyses were conducted by atomic absorption and/or atomic emission [30], following the analytical methods described by Pelkin-Elmer [31]. Mineral content was expressed both as mg/100 g of fresh tissue or as % dry matter (DM).

Cholesterol content was determined by duplicate by gas chromatography according to Fletouris et al. [32]. The fatty acid composition was determined by direct fatty acid methyl esters (FAME) synthesis as described by Cantelolops et al. [33]. The FAME was analyzed by an Aligent 6890 GC system (Aligent Technologies, Santa Clara, CA, USA) equipped with a flame ionization detector and capillary column CP-sil388 (30 m length, 0.25 mm i.d., 0.20 μm film thickness) with a split injection of 1:50. Helium was used as a carrier gas. The temperature of the detector and injector was 250 °C. The initial temperature in the oven was 100 °C, and it reached 220 °C with an increasing rate of 5 °C/min. The fatty acids were identified by comparing their FAME retention times with sigma reference standards (Supelco™ 37 Component FAME mix, Sigma, St. Louis, MO, USA). FAME mix contains n3, n6, and n9 isomers. Results were reported as g/100 g of fresh muscle tissue.

### 2.4. Atherogenic (AI) and Thrombogenic Indexes (TI) and H/h Index

The risk of atherosclerosis and/or thrombogenesis was evaluated by using the atherogenic (AI) and thrombogenic indexes (TI). They were calculated based on the obtained fatty acid results, using the following equations [34]: AI = (C12:0 + 4 × C14:0 + C16:0)/[ΣMUFA + Σ (n − 6) + (Σ (n − 3)]; TI = (C14:0 + C16:0 + C18:0)/[0.5 × ΣMUFA + 0.5 × Σ(n − 6) + 3 × Σ(n − 3) + Σ(n − 3)/Σ(n − 6)]. The ratio between hypercholesterolemic (H) and hypocholesterolemic fatty acids was calculated as described by Monteiro et al. [35]: H/h = (C14:0 + C16:0)/ (C18:1 + C18:2 + C18:3 + C20:3 + C20:4 + C20:5 + C22:4 + C22:5 + C22:6).

### 2.5. Statistical Analysis

A One-way Analysis of Variance (ANOVA) was performed using a mixed model with SWA treatment as the main factor and animal as the random effect. The value *p ≤* 0.05 was used to declare the significant difference between the average scores. Tukey’s multiple comparison test was used for the comparison of means. The Bonferroni correction was also performed to adjusts probability of *p*-values. χ² test was used for taste preference data.

## 3. Results

### 3.1. Meat Quality Traits

Meat quality traits evaluated in the *longissimus lumborum* (LL) muscle of fattening pigs fed with different levels of SWA are presented in Table 2. The pH values (measured 45 min and 24 h *postmortem*) for every treatment were in the range of normal values without being statistically different among treatments (*p* > 0.05). The ANOVA detected a significant effect of treatment (*p* = 0.05) on the a value (redness). Meat samples from the 4%-SWA treatment had less red intensity values when compared to those from the 2%-SWA and the control group (*p* ≤ 0.05). No difference (*p* > 0.05) was observed for L value (lightness) and b value (yellowness). The instrumental tenderness and the cooking losses (expressed in percentage) were not affected by the SWA treatments (*p* > 0.05). The samples from the control group and 4%-SWA resulted in the best taste preference by panelists, being significantly different from the sample of the 2%-SWA group (*p* ≤ 0.05). Samples from the control and the 4%-SWA groups showed a similar percentage of taste preference (Appendix A).

### 3.2. Nutrient Composition of Meat

The ANOVA revealed that the SWA treatments only affected the total content of ash (*p* < 0.001). The LL samples of animals that were fed with the 4%-SWA presented a higher percentage of total ash compared to the other treatments (Table 3; *p* ≤ 0.05). However, no mean differences in the total ash content were found (*p* > 0.05) when comparing the control with the 2%-SWA group. The LL samples from the 4%-WSA exhibited a decreased percentage of total lipids without being statistically different (*p* = 0.07).

The ANOVA showed a significant effect of the SWA treatments on the total cholesterol content (*p* ≤ 0.05). Figure 1 shows that meat samples from animals fed with the highest % of SWA (4%) contained 3.37 and 3.81 mg/100 g more muscular tissue cholesterol content than the control and the 2%-SWA groups, respectively (*p* ≤ 0.05).

The ANOVA showed that the SWA inclusion in the diet of fattening pigs did not affect the composition of saturated (SFA), monounsaturated (MUFA), and polyunsaturated fatty acids (PUFA) detected in the pork meat samples of this study. The ANOVA also found that the SWA treatments did not affect (*p* > 0.05) any of the health indexes associated with fatty acids composition: the ratio hypercholesterolemic index/hypocholesterolemic index, atherogenic index, thrombogenic index. In Appendix A are shown the descriptive statistics for fatty acids composition and health index in pork LL muscles.

Table 4 shows the mean values and standard error mean of several minerals evaluated in the LL samples of this study. The SWA treatments affected (*p* ≤ 0.05) the content of the micro-minerals Mn, Fe, and Cu. The samples of meat with the greatest percentage of SWA (4%) had 0.13, 0.45, and 0.23 less mg/100 g of muscle in Mn, Fe, and Zn, respectively, compared with the control group (*p* ≤ 0.05). The content of minerals of Na Mg and Zn were not different (*p* > 0.05) among the treatment groups; however, there was a trend (*p* = 0.08) of meat samples from the 4%-WSA treatment containing an inferior amount of K when compared to the 2%SWA and the control groups.

By expressing the proximal composition of the meat on a dry basis (DM), the highly significant effect on the ash content was confirmed (Table 5; *p* < 0.0001). The samples from the animals that consumed the highest amount of SWA administered in the pigs’ diet generated an increase in the amount of total ash DM of the meat. The Mn and Cu content decreased (*p* = 0.002) in the treatment with 4%-SWA, compared to control samples, but they were statistically similar to those of the 2%-SWA treatment. In addition, a trend could be evidenced in the results obtained in the Fe content (*p* = 0.06), where meat samples from 2%-SWA exhibited the lowest content of this mineral.

## 4. Discussion

The effects of the inclusion of additives or supplements based on brown seaweed have been tested on growth performance, nutrients digestibility, prebiotic, antioxidant, anti-inflammatory, and immunomodulatory activities in pigs [21,22,23,24]; however, the impact of these dietary interventions on pork meat quality and nutritional composition has been less studied.

The main traits that define pork quality are pH, color, and water holding capacity [36]. The *postmortem* pH variation is an important factor that determines meat quality and has an influence on the physicochemical traits and shelf life [37,38]. Former investigations had found that the addition of seaweed in the animal diet could change carcass characteristics like marbling, color, and pH in pork [25,26,27,39]. Muscle pH values in this study were not affected by the inclusion of SWA and were ranged in the normal values. Muscular pH is highly correlated with the energy content of the diet [40]; however, the rate of pH decrease is influenced by multiple *antemortem* factors [41] and *postmortem* manipulation [38,42]. Michalak et al. [22] reported no statistically significant effect of a green seaweed additive (*Enteromorpha* sp.) on pH, water capacity holding, and drip loss in pork meat. Rossi et al. [19] found that sensory traits like aroma, flavor, and aroma of rabbit meat were affected by the use of 0.3 and 0.6% of dietary brown seaweed in the diet (*Laminaria* sp.). In this study, a better taste preference was observed from pork samples of animals that were fed with the highest % of SWA, without being different from those from the control groups (Appendix A).

According to the pork meat color standard [43], lightness values between 37 and 49 are considered normal. In this study, L values were not affected by treatments, and their values ranged from 44.95 to 47.61. Brown seaweed has been reported to be a rich source of natural antioxidants such as polysaccharides and polyphenols, which could improve meat color display [28,44]; also, antioxidant, antimicrobial, and immunomodulatory activities have been reported for compounds (extracts) of brown algae [8]. Moroney et al. [25] reported that spray-dried seaweed extracts that contained laminarin and fucoidan did not affect the redness values when incorporated in fresh pork via the animal diet. However, in another study, the same authors [26] reported that the SWA significantly reduced the redness intensity compared with the control. Additionally, Rajauria et al. [28] reported that an addition of 5.3% of seaweed extracts (*Laminaria* spp.) in the diet of finishing pigs significantly reduced the redness intensity compared to the control. In this study, the redness intensity was also significantly reduced in samples from animals that were fed with the highest percentage of SWA (4%).

The lower values detected in meat samples from animals that were fed with the highest % of SWA in this study could have been related to the lower iron content present in the muscle from the same treatment (4%-SWA), or could also be related to some interactions between the polysaccharides present in the SWA and the oxymyoglobin in the pork meat [14].

In this study, meat samples from 4%-SWA treatments had lower levels of iron. Ponnampalam et al. [45] reported that increased muscle heme iron concentration resulted in higher values in beef displayed for 48 to 72 h *postmortem*.

We hypothesized that samples from pigs that were fed with the highest % of SWA would increase the PUFA values and reduced cholesterol content; however, meat samples from animals that were fed a regular diet plus 4% SWA got the highest levels of total cholesterol compared to the other treatments (*p* < 0.05). Supplementation of laying hens with 1–2% of dried *Enteromorpha porifera* seaweed resulted in a reduction in cholesterol in the yolk [18]. Rossi et al. [19] reported that the inclusion of dietary levels of brown seaweed (*Laminaria* spp.) did not affect the content of cholesterol in rabbit meat. To our knowledge, there is no previous report on the effect of dietary SWA on cholesterol content in pork samples. Our results suggest that there is an important effect of the components of seaweed that impact the content of cholesterol. Muscle samples from pigs that were fed with 2%-SWA exhibited the highest % of total lipids (5.26% DM) with no significant differences with the other groups. It would be important to confirm these results with a carcass with similar intramuscular content since it is known that there is a high correlation between intramuscular fat and meat lipid composition [46]. Ruqqia et al. [47] tested 13 seaweed extract from different species for hypolipidaemic potential in normal rats, and they found that some of these extracts caused a decrease in total serum cholesterol, triglyceride, and LDL cholesterol but an increase in HDL cholesterol. These results suggest that not all seaweed has the same effect on cholesterol metabolism. Since seaweed represents a group of organisms with diverse types of bioactive compounds, further studies are needed to understand the biological effects of the extract of *Macrocystis pyrifera* on cholesterol metabolism.

The proximal composition of the SWA used in this study is described in Table 1. Lipid content, crude protein, and ash content are lower than the values reported by Ortiz et al. [14] in the fresh seaweed (*Macrocystis piryfera*). These authors reported (in DM) 10.8% for ash, 13.2% for crude protein and 0.7% for total lipids, and 75.5% for carbohydrates. Seaweed composition depends on the harvest conditions, the habitat, and many other external conditions such as water temperature, light intensity, and nutrient concentration in the water [48].

The fatty acid composition of intramuscular tissue is affected by dietary lipid composition, de novo lipogenesis, desaturation, and the difference in the utilization of various fatty acids by the animal body [49]. In LL samples from this study, palmitic acid (C16:0) was the most abundant SFA, oleic acid (C18:1*c*
*n*−9) was the most abundant MUFA, and linoleic acid (C18:2 *n*−6) the most abundant PUFA in the LL of the examined pigs. The composition of the fatty acids of the LL samples from this study is similar to those reported by Alonso et al. [50] and Parunovic et al. [51] in pork meat with no dietary seaweed inclusion. Comparing the fatty acids composition of LL samples with *Macrocystis pyrifera* seaweeds, the most abundant MUFA was also 18:1*c n*−9 (oleic acid) with 19.64 ± 0.08 %; the linoleic acid (18:2 *n*−6) reached values of 43.41% and the predominant SFA was also palmitic acid (C16:0; 16.17 ± 0.06) [14]. El Bahr et al. [52] reported a significant increase in the levels of EPA, DHA, total PUFA, and arachidonic acid in breast muscle of broiler chickens fed with microalgae extracts (1 g/kg diet), suggesting that high contents of methionine and lysine (present in the microalgae) were positively correlated with the increase in PUFA. *Macrocystis pyrifera* contains low contents of proteins and essential amino acids like methionine and lysine [14] compared to a microalgae, like *Arthrospira* sp. [27].

Several studies found that the long-chain ω-3 PUFA content in the muscle or adipose tissue was largely independent on the timing of feeding ω-3-PUFA-rich diets [53]; in this study, pigs were fed with the SWA for 45 days during the fattening phase; however, the effect of dietary SWA was not significant. Perhaps, it is necessary to incorporate the SWA during the growing and/or finishing phase to evaluate the impact on meat chemical composition.

Fat and fatty acids are important because of their effects on human health. In this study, the index value H/h and IA ratios turned out to be less than 1 (Appendix A), which indicated that regardless of the inclusion of SWA in the diets, all samples are categorized as healthy meats. The relation between H/h comes from the functional effects of fatty acids in cholesterol metabolism and gives a superior measure of the nutritional evaluation of fats from a nutritional standpoint [35]. The relation H/h is a suitable indicator to evaluate the risk of elevated blood cholesterol since it excludes C18:0 but includes two important hypercholesterolemic, palmitic acid (C14:0) and oleic acid (C16:0) are known to be the most important hypercholesterolemic fatty acids [35].

The balance in the relationship ω6 and ω3 (ω6/ω3) plays an important prevention role for severe chronic disorders and autoimmune diseases, and these authors agree to the average recommended value of 5:1. The ω6/ω3 ratio, which is currently recommended by the OMS, should be lower than 10, in *M. piryfera* this ratio is 7.42 [54].

The inclusion of dietary SWA affected the content of ash; however, in this study, microminerals such as Mn, Fe, and Cu were found in less quantity in meat samples from animals that consumed the greatest proportion of SWA (4%). All seaweeds are characterized by a higher ash content (19.3–27.8% DM) than those observed in edible plants [21], being considered by some authors [27] as an important organic source of minerals for livestock nutrition; however, there is wide variability in mineral content among seaweed species. *Macrosysty pyrifera*, for example, is rich in Mg (39 ± 2.8 mg/g), Na (36.9 ± 9.9 mg/g), K (67.5 ± 22.3 mg/g) and Fe (117 mg/kg) [11,14]. On the other hand, these algae have a relatively low content of Mn (11 mg/kg), Zn (12 mg/kg), and Cu (2 mg/kg) compared to other species of brown seaweed [11,21]. There is no scientific evidence of the direct relationship between the mineral content of the dietary seaweed additive and the mineral content in meat. The numerous bioactive compounds that are present in fed additive could affect the chemical composition of meat. Moreover, the content of polysaccharides such as alginates and agar or carrageenan could cause the formation of insoluble complexes with minerals, decreasing their bioavailability [55].

It has also been reported that *Laminaria* spp. is rich in alginates, which probably hampers the bioavailability of Ca, and that the apparent absorption values of Na and K were significantly higher in rats supplemented with *Laminaria* spp. while Mg absorption was not affected [56]. Several components in fed matrices can also exhibit retention properties in minerals, such as phenolic compounds and phytic acid, which reduce the bioavailability of Fe and Zn.

## 5. Conclusions

The meat of pigs fed with brown sea algae additives had a less intense red color than the control pigs. From a nutritional point of view, the meat of pigs fed with a higher percentage of seaweed additive (4%) had a small but significant total ash increase and a lower percentage of total lipids. However, the fatty acid composition of pig meat was not influenced by including seaweed additive, and some microminerals like Cu, Zn, and Mn decreased in content. In general, no harmful effects were found in this study by feeding pigs with brown sea algae extracts. Since seaweed represents a group of organisms with a diverse type of bioactive compounds, further studies are needed to understand the biological effects of this SWA on adipogenesis, cholesterol metabolism, and mineral deposition in muscle.

## Figures and Tables

**Figure 1 foods-10-01720-f001:**
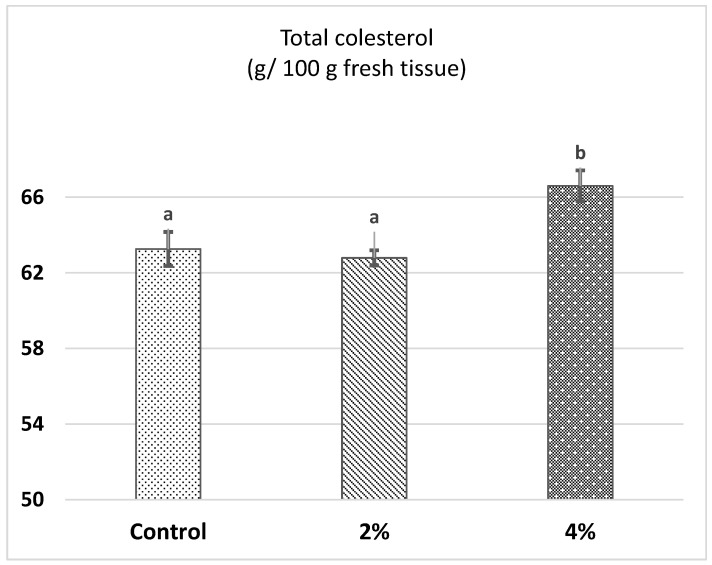
Mean values ± standard error means of total cholesterol in pork meat according to SWA treatments (*p* = 0.02). The different superscript letters represent significant statistical differences (*p* ≤ 0.05).

**Table 1 foods-10-01720-t001:** Composition of the regular diet and chemical composition of the seaweed additive (SWA).

Diet Ingredient	%	Chemical Composition of the Seaweed Additive	% *
Triticale	70.00	Dry matter	92.67
Wheat bran	8.10	Ash	25.60
Soya meal	20.20	Crude protein	8.87
Salt	0.42	Crude fibre	2.87
Calcium carbonate	0.60	Neutral-detergent fibre	7.06
Phosphate	0.17	Acid-detergent fibre	11.78
Oil	1.01	Ether extract	0.29
Vitamin-mineral premix ^1^	0.10	Nitrogen free extract	55.04
Lisin	0.08		
Methionine	0.04		
Threonine	0.02		
quantum blue ^2^	0.01		
Enocase ^3^	0.02		

* based on dry matter. ^1^: contains vitamins: A, D3, E, K3, B2, and B12; and minerals: Mn, Cu, I, Zn, Fe, Se, and Ca. ^2^: an enhanced E. coli phytase. ^3^: an enzyme preparation with endo-1,4-β-xylanase.

**Table 2 foods-10-01720-t002:** Effects of the inclusion of seaweed additive (SWA) in the diet of fattening pigs on meat quality traits.

Variable	Treatment SWA			SEM	*p* Value
	Control	2%	4%		
Muscular pH, 45 min	6.08	6.34	6.21	0.08	0.13
Muscular pH, 24 h	5.61	5.64	5.63	0.07	0.93
Redness (a value)	5.88 ^a^	5.61 ^a^	4.95 ^b^	0.24	0.05
Yellowness (b value)	9.32	9.60	9.06	0.14	0.18
Lightness (L value)	45.79	47.61	4495	0.96	0.37
Cooking loss, %	17.15	16.55	14.85	0.42	0.23
Shear force, kg	2.27	2.22	2.11	0.08	0.86

Means within a row lacking a common superscript letter differ (*p* ≤ 0.05). SEM: Standard Error Mean.

**Table 3 foods-10-01720-t003:** Effects of the inclusion of seaweed additive (SWA) in the diet of fattening pigs on the proximate composition of pork meat.

Variable ^1^	Treatment SWA			SEM	*p* Value
	Control	2%	4%		
Moisture	73.77	74.08	74.47	0.21	0.26
Dry matter	26.92	25.92	25.52	0.26	0.26
Total ash	1.23 ^a^	1.26 ^a^	1.42 ^b^	0.01	0.0001
Crude proteins	23.77	23.24	23.13	0.17	0.46
Total lipids	1.13	1.36	1.05	0.12	0.07

^1^ values are expressed as g/100 g of fresh muscular tissue. Means within a row lacking a common superscript letter differ (*p* ≤ 0.05). SEM: Standard Error Mean.

**Table 4 foods-10-01720-t004:** Effects of the inclusion of seaweed additive (SWA) in the diet of fattening pigs on the mineral content in pork meat.

Mineral Content	Treatment SWA			SEM	*p* Value
	Control	2%	4%		
Macrominerals ^1^					
Na	44.57	50.15	44.35	1.82	0.20
Mg	23.12	25.28	24.75	1.29	0.71
K	492.67	504.62	430.03	12.50	0.08
Microminerals ^1^					
Mn	0.22 ^b^	0.16 ^ab^	0.09 ^a^	0.01	0.002
Fe	1.75 ^b^	1.18 ^a^	1.30 ^ab^	0.08	0.04
Cu	0.42 ^b^	0.36 ^b^	0.19 ^a^	0.02	0.001
Zn	1.31	1.41	1.26	0.04	0.35

^1^ values are expressed as mg/100 g of fresh muscular tissue. ^2^ values are expressed as mg/g of fresh muscular tissue. Means within a row lacking a common superscript letter differ (*p* ≤ 0.05). SEM: Standard Error Mean.

**Table 5 foods-10-01720-t005:** Means of proximate composition and mineral content based on dry matter (DM) by treatments.

Variable	Treatment SWA			SEM	*p* Value
	Control	2%	4%		
Proximal composition ^1^					
Ash ^1^	4.71 ^a^	4.87 ^a^	5.55 ^b^	0.12	<0.0001
Crude protein ^1^	90.64	89.68	90.64	0.98	0.53
Total lipids ^1^	4.31	5.26	4.13	0.91	0.43
Macrominerals ^1^					
Na ^1^	0.17	0.19	0.17	0.02	0.31
Mg ^1^	0.088	0.096	0.097	0.001	0.68
K ^1^	1.87	1.92	1.66	0.11	0.11
Microminerals ^2^					
Mn ^2^	8.30 ^b^	6.06 ^ab^	3.82 ^a^	0.96	0.002
Fe	66.90	45.06	50.38	2.64	0.06
Cu	15.93 ^b^	13.64 ^b^	7.72 ^a^	1.69	0.004
Zn	50.32	53.94	48.82	4.42	0.51

^1^ values expressed as g/100 g of DM. ^2^ values expressed as mg/kg of DM. Means within a row lacking a common superscript letter differ (*p* ≤ 0.05). SEM: Standard Error Mean.

## Data Availability

Data are not available in public datasets, please contact the authors.

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
