# Peer review of "Effect of Dietary Brown Seaweed (Macrocystis pyrifera) Additive on Meat Quality and Nutrient Composition of Fattening Pigs"

_foods, 2021, doi:10.3390/foods10081720_

Round 1

Reviewer 1 Report

Report on the manuscript foods-1228295 entitled: Effect of dietary brown seaweed (Macrocystis pyrifera) additive on meat quality and nutritional composition of finisher pigs.

Major comments:

  • According to the title, finishing (not finisher) pigs were considered in the study. Nevertheless, in L. 81 and 82, it is said “approx. 14 weeks of age” and “before fattening phase”. Finally, in L. 90, “slaughter size 95 kg” was stated.
    Could the authors provide more information about the considered production system? I am not sure if the stage was really “finishing” or was actually “fattening”… (growing, growing-fattening, fattening, finishing). 95 kg could be considered as a low weight for slaughter.
  • First of all, the composition of the SWA additive, regarding FAs, Aas, minerals… has not been included. Therefore, it is difficult to review (and validate) the results from Tables 3 to 6 and figure 1.
    In the introduction, it is described that, although no differences in protein amount have described in the literature, differences in FAs, AAs and minerals are commonly found on SWA.
    The authors conclude that the dietary inclusion of 2 and/or 4% of SWA caused minimal or none changes. Then… why was such additive considered?
    If minimal or none effects were detected, why should further studies need to be done?
    Reviewing the mean and SEM values from all the tables, the authors must consider the statistical analysis of their results but using a less conservative tests such as Duncan or Bonferroni (instead of Tukey) in order to describe the modest observed changes.
    By improving the results and discussion sections, the manuscript will increase its novelty, quality, scientific soundness and interest.

Minor comments:

  • L. 54 and 58. Reference 12 is used but, in that reference, seaweed Sargassum (Sargassaceae) is described, not Macrocystis pyrifera. Please, explain.
  • L. 59. In reference 16, a description of the different seaweeds and the compounds that can be extracted from them is described. I cannot see the relationship with “beef cattle”. Please, explain.
  • L. 60. Reference 19 is about biochemistry of FgPLD genes, not related to “laying hens”. Please, explain.
  • L. 61-62. This issue deserves a bit more attention. Reference 22 is ok but there are in the literature more interesting references to describe such issue. Please, add more information and substitute ref 11 and 21.
  • L. 67-70. No studies can be found about pigs/pork? Your are using references from a complete different specie (monogastric vs ruminant/polygastric)…
  • L. 71-74. This should be previously justified! (Please, see comments above).
  • Table 1: Please, describe in the footnote the composition of the vitamin mineral premix and the quantum blue. Could the authors explain what “enocase” is?
  • L. 104. Immediately or 48 h post-mortem?? Which one?
  • L. 137. Hydrogen as a carrier gas? Is this right? Usually, the carrier gas is He or N2. H2 flame heats the system.
  • L. 147-148. AGMI? Meaning?
  • L. 151. Which isomers? N3, n6, n7, n9?

Reviewer 2 Report

COMMENTS FOODS 1225295

Effect of dietary brown seaweed (Macrocystis pyrifera) additive 2 on meat quality and nutritional composition of finisher pigs

 WORK’S STRENGTHS

The work points out a use of natural source to feeding pigs, the from that point of view, the work is interesting, providing some information about meat quality and nutritional value of pig meat.

WORK’S WEAKNESS

Although the experimental design can be considered interesting, the work has not the scientific level to worth be published in a high impact journal as Foods.

SPECIFIC COMMENTS

Abstract

The abstract is concise and understandable itself.

Introduction

The introduction section is not enough documented but properly organised.

Materials and Methods

2.1. This section is properly described.

Results

  • The authors indicate treatment differences in several parameters, with no significant differences. I am aware that sometimes import differences in absolute values did not be reflected in statistically differences because of the high standard deviations. However, this is not the case, because the absolute values only differ in a small percentage of absolute (ie. Only 13% in C18:2 and 13% in PUFA content) and P value far to be significant P=0.4 or P=0.7 respectively.

Discussion

  • The discussion is too long and deep in parameters which this study has not report an experimental treatment effect.
  • In the most of parameters the discussion seems more a summarize of general knowledge that a proper discussion to results obtained in the work.
  • Seaweed addition only have important effect in mineral contents, and in these results are better discussed.

Conclusion

This conclusion summarizes properly the results.

Round 2

Reviewer 2 Report

The manuscript has been improved, since the introduction and discussion section has now more scientific soundness.

However, in my opinion, a scientific paper in a high impact journal as Foods, must include more parameters to study than nutritional value and colour. I mean the need of taking muscle sample to develop some other important parameters in meat quality such as those related to sensory quality, at least in some of animal tested. In this sense, it is known that some supplement in feed livestock animals, in spite of improve their nutritional value, the final product is not suitable to be commercialize and, actually, is not profitable to use for producers.

The authors stated that “any research team can build new research questions by considering previous results”. I agree, but in these cases, these previous results are not enough to be published as a scientific paper until more complete and conclude and studies have been obtained.

This is just my recommendation, but of course, the final decision is dependent of the editor criteria.

Author Response

We appreciate the reviewers’ work on providing ideas, constructive critics, and valuable suggestions. We also accept your opinion about our work; however, we think that this study contributes to providing information about this new additive which is a lyophilized product of Macrocystis pyrifera (brown seaweed widely found in the Chilean coasts). The results found in this study had proved that the additive had no detrimental effect on meat quality traits; however, it caused variation in the ash content, cholesterol, and some micro minerals of pork meat. Also, in applied research, not all results end up with a positive recommendation for a potential user (in this case to producers). We are confident that a new finding deserves to be published, in this case, our research (according to reviewers) does not have experimental design flops that could question the results obtained.

However, your comments allow us to decide the inclusion of preliminary results of the sensory analysis that we had performed. This sensory analysis was about only taste preference, which at the beginning we did not include because we consider it was necessary to increase the number of replicates. We used two steaks of each treatment in two sessions. The test was carried out in individual evaluation cabinets illuminated with red light, using 19 panelists (16 women and 3 men). Each panelist, in each session, tasted 3 samples (one from each treatment) at random and they were asked to select the best preference of the three samples. We have decided to include the results of this sensory analysis for taste preference, which are available as Supplementary Material. With this information, we could answer the question of whether or not the inclusion of dietary SWA in fattening pigs could affect pork meat flavor.